# Physician burnout in the context of the COVID-19 pandemic: prevalence and associated factors among resident doctors and consultants in Delta State, Nigeria

Nnamdi Stephen Moeteke[1,2]*, Ezinneamaka Erhirhie[3,4]

1 Department of Community Medicine, Delta State University Teaching Hospital, Oghara, Nigeria,
2 Department of Community and Public Health, Idaho State University, Pocatello, Idaho, United States of America, 3 Department of Family Medicine, Delta State University Teaching Hospital, Oghara, Nigeria, 4 Mersey and West Lancashire Teaching Hospitals NHS Trust, Merseyside, United Kingdom

*nnamdimoeteke@gmail.com

## Abstract

Burnout is a state of emotional, physical, and mental exhaustion caused by long-standing, poorly managed workplace stress. Resident doctors (RDs) and consultants provide specialized medical care, and the strain of their job predisposes them to the three domains of burnout: Emotional Exhaustion (EE), Depersonalization (DP), and diminished Personal Accomplishment (PA). Globally, this public health crisis worsened with the overwhelming effect of COVID-19 on health systems. This study assessed the prevalence and associated factors of burnout among RDs and Consultants in tertiary hospitals in Delta State, Nigeria, during the pandemic. A cross-sectional design was employed. Previously validated instruments (the Maslach Burnout Toolkit, and the Pandemic Experience and Perception Survey) were used to collect data via an online survey. The questionnaire was sent to physicians selected by a multistage sampling. The proportion of participants with a high grade in each of the domains of burnout was obtained. Stepwise analyses from bivariate to multivariate were done to obtain adjusted odds ratios. The prevalence of high-grade burnout in EE, DP, and PA was 35.1%, 13.2%, and 33.3% respectively. Relative to those ≤ 30 years, the age group 41 – 50 years had less likelihood of high EE (AOR 0.050; 95% CI 0.004 – 0.651). Other independent predictors of EE were manageable workload (AOR 0.094; 95% CI 0.027 – 0.328), reward for work (AOR 0.427; 95% CI 0.205 – 0.892), and good leadership (AOR 0.525; 95% CI 0.113 – 0.929). This study suggests that the determinants of burnout among RDs and consultants are mainly contextual factors in the work setting. Future interventions to control physician burnout in Nigerian healthcare systems should be geared towards promoting an institutional culture of leadership, manageable workloads, and appropriate rewards for good performance.

**Data availability statement:** All relevant data are within the paper and its Supporting Information files.

**Funding:** The author(s) received no specific funding for this work.

**Competing interests:** The authors have declared that no competing interests exist.

## Introduction

Medical practice and its demands of saving lives can be psychologically stressful [1]. Long on-the-job hours, including call duties, are the norm[2]. These strains are amplified in developing countries like Nigeria, with constant organizational challenges and resource shortages, which lead to incessant brain drain and more workload for the remaining manpower[2,3]. These predispose physicians and other healthcare professionals to burnout. Burnout is characterized by a combination of associated symptoms emanating from long-standing poorly managed workplace stress, with three facets, namely:

a) Emotional exhaustion (EE)—feelings of being drained or energy-depleted

b) Depersonalization (DP)—increased mental and emotional distance from one's job, or feelings of negativism and cynicism towards one's job

c) Diminished personal accomplishment (PA)—reduced professional self-esteem or feelings of low success [4].

Globally, there was an increase in burnout among health professionals during the COVID-19 pandemic, with many having to work in overwhelmed health systems, associated with high levels of risk, anxiety, and mortality[5]. The impact of burnout on physicians includes depression and poor general health, individual and work-place impairment, substance abuse, and suicide. These ultimately affect the quality of care of patients (through decreased professional capacity, indifference and loss of empathy for patients, and higher chances of healthcare errors), the health organizations/systems (absenteeism and high turnover of already trained personnel), and colleagues/families of affected doctors (relationship friction), making it a 'public health crisis' [6–9]. A survey conducted in Eastern Nigeria just before the pandemic got to the country revealed that the prevalence of burnout was 69% among healthcare workers, with as high as 40% screening positively for depression, and the domains of burnout correlating strongly with decreased productivity[10]. Therefore, the pressure on Nigerian healthcare systems due to COVID-19 exacerbated the challenges already faced by physicians in the country. The latest 'Physicians' Oath' acknowledges the welfare of the medical doctor, and inductees are now required to swear to "attend to my own health, well-being, and abilities in order to provide care of the highest standard"[11]. However, unlike somatic ailments, smoldering symptoms of burnout are often put on the back burner, especially in developing countries where the condition is under-reported[12]. Though the term "burnout" has been around since the 1970s, it is acknowledged that more data is continually required to design evidence-based control measures among physicians[1].

The Maslach Burnout Inventory (MBI), a 22-item questionnaire, is the gold standard and the most prominent tool for measuring burnout as defined by the World Health Organization (WHO) [9,13]. It has been validated by countless research on burnout that have been done since the tool was developed over four decades ago[14]. MBI measures burnout in each of the three dimensions using questions with responses on a 7-point scale: 'Never', 'a few times a year', 'once a month',

'a few times per month', 'once a week', 'a few times a week', and 'every day,' with the scores ranging from 0 - 6. There are nine questions for 'Emotional Exhaustion' (EE), five for 'Depersonalization' (DP), and eight for 'Personal Accomplishment'(PA) [15]. In the original MBI Manual [16], scores in each domain are categorized into "low," "moderate," and "high" burnout (Table 1). The three dimensions are non-cumulative.

The absence of a universal interpretation of the MBI whereby overall burnout is presented as a single binary variable is problematic. While some investigators describe overall burnout as the presence of high burnout in all three domains, others have considered study participants to be truly burnt out if at least two sub-scales are in the 'high' category [17,18]. Other authors see burnout as having either high emotional exhaustion or high depersonalization [19]. Overall burnout has also been estimated based on participants having a high burnout in at least one of the three domains[15] These variations make it difficult to compare studies, with some studies whose authors belong to the first two schools of thought profoundly underestimating rates, at least potentially[20]. While the developer of the instrument supports the idea of defining clinical burnout as the presence of a high EE burnout alongside either a high DP or PA burnout, she recommends that when assessing for associations between burnout and other variables, it is best to use individual subscale scores as numerical data, or, less preferably, to report burnout in each domain as a categorical variable based on the already established cut-offs for 'high', 'moderate', and 'low' [20].

In Nigeria, resident doctors (RDs) are registered medical/dental graduates working and undergoing specialized training (under the supervision of consultants) in health facilities (mostly tertiary hospitals) accredited by the regulatory bodies for graduate medical education[21]. RDs begin their training as Registrars, and are promoted to the rank of Senior Registrar when they pass the Part 1 Fellowship Examination for their specialty which is usually taken after a minimum of three years. A consultant is a medical/dental practitioner who, having completed the residency training and passed the Part 2 Fellowship Examination, works as a specialist in a health facility or medical school [22]. There are about 17,000 RDs—about 40% of the doctor population—and over 4000 consultants in Nigeria[23,24]. It is difficult to ascertain the prevalence of overall burnout among physicians in low- and low-middle-income countries due to variations in research instruments and criteria. Nine out of the studies obtained from literature search employed the MBI and the true prevalence of high EE, DP, and PA burnout may only be estimated from these studies. The papers that examined both RDs and consultants, like this current study, reported levels of high EE, DP, and PA burnout ranging from 20% - 39.4%, 30% - 71%, and 25% - 50.3%, respectively [25–28].

The relationship between socio-demographic characteristics and physician burnout is unclear as the evidence varies[15,25,29–31]. However, the higher likelihood of burnout among younger physicians and those with fewer years in practice has been highlighted[25,26,29,32]. More consistent predictors of physician burnout are work- and

**Table 1. Categorization of MBI Scores [16].**

| Subscale | Category | Cut-off Scores |
|---|---|---|
| Emotional exhaustion (Score: 0 – 54) | High | ≥ 27 |
| | Moderate | 19 – 26 |
| | Low | 0 – 18 |
| | | |
| Depersonalization (Score: 0 – 30) | High | ≥ 10 |
| | Moderate | 6 - 9 |
| | Low | 0 - 5 |
| | | |
| Diminished personal accomplishment (Score: 0 – 48) | High | 0 - 33 |
| | Moderate | 34 - 39 |
| | Low | ≥40 |

training-related factors such as frequency/duration of being on call, number of work hours/workload, remuneration/economic problems, organizational support, recognition from hospital managers, getting adequate skill development and professional training including pandemic-related training, satisfaction with hospital infection control measures, and frequency in dealing with suspected, confirmed, and critical cases of COVID-19. These indicate that determinants of burnout are mostly contextual factors related to organizational arrangements and management systems in the work setting [33]. This is corroborated by the qualitative aspect of the study by Ghazanfar et al [34]. In low- and low-middle-income countries, only five studies done in Egypt (2), Ethiopia, Pakistan, and Sri Lanka explored, apart from workload, different aspects of these institutional predictors, albeit superficially[25,30,32,34,35]. Furthermore, only two studies done in Egypt provide data on the prevalence and associated factors of physician burnout during the COVID-19 pandemic[25,32]. There is a paucity of studies in Nigeria involving both residents and consultants, assessing organizational factors and providing data on physician burnout in the context of the pandemic. This study sought to fill the gap in the literature as regards physician burnout during the COVID-19 pandemic and highlight critical areas of strength and or weakness in healthcare organizations to guide prevention and control strategies. Therefore, the aim of this study was to assess the prevalence of burnout among RDs and Consultants working in tertiary health facilities in Delta State, Nigeria, and determine the associated factors such as socio-demographic characteristics, features of the work setting, and experiences/perceptions of work during the pandemic.

## Methods and materials

### Ethics statement

Ethical clearance was obtained from the Health Research Ethics Committee of Delta State University Teaching Hospital (Approval Number HREC/PAN/2021/037/0426). The licenses to use the Maslach Burnout Toolkit and the Pandemic Experience and Perceptions Survey (PEPS) were purchased from the copyright holder, Mind Garden, Inc. The study participants had to go through the Participant Information Sheet (PIS) and consent statements after clicking on the link to the online survey. They were informed that they could withdraw from the study without repercussions and were required to tick checkboxes indicating their consent before they could proceed to answer the questionnaire. The PIS contained the researchers' telephone numbers and email addresses to enable potential participants to ask questions if needed. It also had contact details of support groups/organizations that participants could contact if the questionnaire caused emotional distress. There were no identifiers on the questionnaire, and the online survey had no IP links to ensure anonymity. The data was stored in a password-protected laptop that was accessible to the investigators only. There was no incentive or compensation for participants.

### Study design

A cross-sectional design, which gives a 'snapshot' of an outcome of interest in a population, was used. This was most appropriate since it is the only design that can both measure the health status of a given population at a specific place and time and determine associated factors for the outcome of interest [36].

### Study setting

This study was done in the two tertiary health facilities in Delta State, Nigeria. These are government-owned hospitals providing the highest level of specialized and critical healthcare to inhabitants of the state and its environs. Each hospital provides services in these specialties: anesthesia, community medicine, emergency medicine, family medicine, internal medicine, obstetrics & gynecology, ophthalmology, orthopedics, otorhinolaryngology, pediatrics, radiology, and the various subspecialties of surgery. There were a total of 502 RDs and Consultants working in both institutions.

The index case of COVID-19 in Nigeria was reported on February 27, 2020, while the first case in Delta State was confirmed on April 7, 2020[37]. The first wave of the pandemic in the country lasted from February to October 2020; the second wave from October 2020 to June 2021; and the third wave from June to November 2021. The fourth wave began in December 2021 and started to flatten in January 2022. The second wave was the largest, with the number of new cases per month reaching a record high of 43,635 in January 2021. Though the fourth wave had the shortest duration, it was also remarkable as new cases increased by 500% in two weeks, and the week of December 20 – 26 witnessed an unprecedented 14,078 new cases per week[38–45].

### Selection criteria

RDs and consultants who had worked in the facilities for ≥ 1 year were included. Supernumerary RDs and consultants hired on temporary or part-time contracts were excluded.

### Sample size calculation and sampling

The formula for obtaining the minimum sample size (n) for identifying a proportion in a population was used:

$$n = 1.96^2 \, P(100 - P)/E^2$$

where
n = minimum sample size
P = proportion of the population with burnout = 75.5%, from a previous study [31]
E = acceptable margin of error (5%)

$$n = 1.96^2 \, x \, 0.755 \, x \, 0.255/0.05^2 \approx 296$$

Since the total number of RDs/Consultants is less than 10,000, the following correction formula was used to obtain a corrected sample size, $n_c$.

$$\frac{n \, Pt}{n + (Pt - 1)}$$

Where n is the uncorrected sample size, and Pt is the total population of RDs/consultants.

$$nc = (296 \times 502)/(296 + [502 - 1]) \approx 187$$

Assuming a non-response rate of 10%, this required sample size was increased by 10% to 206.

The lists of RDs and consultants for both hospitals (with distribution by specialty) were obtained. Based on the total number of RDs and consultants in each hospital, the total number in each specialty and the numbers of RDs and consultants in each specialty, 206 RDs/consultants were selected by multistage sampling technique. This involved stratified sampling in three stages to proportionately allocate participants to the two hospitals, to individual specialties in each hospital, and to the three cadres (registrar, senior registrar, and consultant) in each specialty within each hospital. Participants were then selected by simple random sampling technique (balloting method) for individual cadres in each specialty for the two hospitals.

### Recruitment

The directory of the physicians' association in the state (to which these researchers belong), which is readily available to members, was used. All doctors have access to this document as a shared resource, and these researchers are permitted to send recruitment

invites out to the list with no need for a gatekeeper. The study was advertised in the doctors' WhatsApp groups with a brief description of the purpose, aim/objectives, eligibility criteria, and contact details of the researcher. The potential participants were given the opportunity to contact the researchers if they had any questions and had two weeks to consider their participation. Thereafter, the web link to the online participant information sheet (PIS), consent form, and questionnaire was sent via WhatsApp to the selected individuals, with a friendly reminder after two, four, and six weeks, respectively. Data collection was done from January 13 to March 5, 2022.

### Method of data collection

Transform Survey Hosting [46] was used. The questionnaire (about 25 minutes to complete) consisted of four sections: socio-demographic/personal professional data, the Maslach Burnout Toolkit, and the Pandemic Experience and Perceptions Survey (PEPS). Maslach Burnout Toolkit "combines the Maslach Burnout Inventory (MBI) and the Areas of Worklife Survey (AWS) to measure burnout in the Worklife context" [47].

The AWS, which has proven validity and reliability in different occupational settings, is a companion tool to the MBI and assesses workplace characteristics that could predict the occurrence of burnout[13]. AWS has 28 items (scored on a five-point Likert scale) which assess the perception of 'Workload' (5 items), 'Control' (4), 'Reward' (9), 'Fairness' (6), and 'Organizational Values' (4). The average score for each sub-component was obtained.

The PEPS (35 items scored on a five-point Likert scale) is a powerful tool for assessing employees' experiences and perceptions during a pandemic. The question items are in five categories: 'Extent of Impact' (3 items), 'Adequacy of Resources' (5), 'Risk Perception' (7), 'Worklife' (7), and 'Leadership' (10) [48]. The total score for each category was obtained.

The survey instruments were pretested among 11 RDs and 10 consultants of a tertiary hospital in a neighboring state to ensure suitability for the study population, and they were revised for clarity. The PEPS had to be adapted for this study by slightly modifying the questions to make them specific to the current pandemic. For example, under 'Leadership', the question "How did you experience leadership during the pandemic period" was changed to "How do you experience leadership during the ongoing COVID-19 pandemic." Similarly, among the items for 'Risk Perception', "For the ongoing COVID-19 pandemic, how much risk to your family do you perceive" was used instead of "For the relevant period, how much risk to your family did you perceive."

### Description of the dependent variables

The dependent variables were EE, DP and PA. Burnout was indicated by high category, i.e., EE score ≥ 27, or DP score ≥ 10, or PA score ≤ 33 [9]. Each participant was allocated to 'burnout' or 'no burn out' to create a dichotomous dependent variable for each subscale. The prevalence of burnout was measured by the proportion (percentage) of participants who met the cut-off in at least one subscale.

### Data analyses

The dataset was checked for missing data and there was none for all the listed variables. The Chi-squared test was used to determine if there was an association between categorical independent variables (e.g., age group, marital status) with each domain of burnout. For continuous independent variables (e.g., score for perception of risk), association with scores for each domain of burnout was tested using Pearson correlation. To identify independent predictors of each domain of burnout, all categorical and continuous variables that were statistically significant on bivariate analyses were introduced into a binary logistic regression model. The level of statistical significance (α) was set at 0.05 for all analyses.

### Results

Only 114 people submitted the filled questionnaire, giving a response rate of 55.3%. The average age of respondents was 40 years (Table 2). Of the 114 respondents, 40 (35.1%) had high EE, 15 (13.2%) had high DP, and 38 (33.3%) had high PA burnout. There was high burnout in at least one domain in 46 respondents (40.4%) (Table 3).

**Table 2. Socio-demographic and professional characteristics of respondents.**

| Characteristic of Respondents | | Total, N = 114 | |
|---|---|---|---|
| | | **n** | **%** |
| Age (years) | ≤30 | 10 | 8.8 |
| | 31 – 40 | 55 | 48.2 |
| | 41 – 50 | 42 | 36.8 |
| | ≥ 51 | 7 | 6.1 |
| | Mean ± SD | 40.30 ± 6.27 | |
| | | | |
| Sex | Male | 66 | 57.9 |
| | Female | 48 | 42.1 |
| | | | |
| Marital status | Single | 11 | 9.6 |
| | Married | 96 | 84.2 |
| | Separated | 3 | 2.6 |
| | Divorced | 2 | 1.8 |
| | Widow | 2 | 1.8 |
| | | | |
| Specialty category | Medical-related | 64 | 56.6 |
| | Surgical-related | 41 | 36.3 |
| | Diagnostic | 8 | 7.1 |
| | | | |
| Rank | Consultant | 22 | 19.3 |
| | Senior resident doctor | 52 | 45.6 |
| | Junior resident doctor | 40 | 35.1 |
| | | | |
| Number of years post- graduation from medical school | ≤10 years | 40 | 35.1 |
| | 11 – 20 years | 58 | 50.9 |
| | 21 – 30 years | 15 | 13.2 |
| | ≥ 31 years | 1 | 0.9 |

The categorical variables significantly associated with EE were age group and number of years post-graduation (Table 4). Only age group was significantly associated with DP (Table 5). EE scores had a weak positive correlation with number of work hours per week, and scores for perception of risk; a weak negative correlation with scores for sense of control, community, fairness, values, and adequacy of resources; and a moderate negative correlation with scores for manageability of workload, reward, and experience of leadership. A similar pattern was obtained for DP scores except that all the correlations were weak (Table 6).

The multivariate model for high EE showed decreasing likelihood of high EE with older age groups, though this was statistically significant only for those aged 41–50 years who had a 95% decrease in odds compared to the group of ≤ 30 years. The AWS and PEPS variables which were independent predictors of high EE were 'manageability of workload', 'reward for work', and 'experience of leadership' (Table 7). There was no statistically significant association between DP and all the predictors entered into the model.

## Discussion

### Implications of the findings

The findings meet the aims of this study, which are to fill the gap in the literature in terms of the prevalence of physician burnout in Nigeria during the pandemic and, beyond socio-demographic characteristics, to explore predictors such as

**Table 3. Distribution of burnout categories among respondents.**

| Burnout domain | | Frequency (%) |
|---|---|---|
| **Emotional Exhaustion (EE)** | Low burnout | 53 (46.5%) |
| | Moderate burnout | 21 (18.4%) |
| | High burnout | 40 (35.1%) |
| | | |
| **Depersonalization (DP)** | Low burnout | 76 (66.7%) |
| | Moderate burnout | 23 (20.1%) |
| | High burnout | 15 (13.2%) |
| | | |
| **Personal Accomplishment (PA)** | Low burnout | 45 (39.5%) |
| | Moderate burnout | 31 (27.2%) |
| | High burnout | 38 (33.3%) |
| | | |
| **Overall Burnout (presence of at least one of High EE, High DP, and High PA burnout** | Yes | 46 (40.4%) |
| | No | 68 (59.6%) |

experiences and perceptions of the pandemic and workplace characteristics. To the best of our knowledge, this is the first study to provide data in this regard in Nigeria.

The results indicate that the prevalence of high burnout in the EE (35.1%) and PA (33.3%) domains, as well as overall burnout (40.4%), is quite elevated among Nigerian physicians who provide specialized and critical care for the populace. However, these levels are slightly less than was obtained in Benin City, Nigeria and Pakistan [15,18], and far less than the other studies in South Africa, Ethiopia, Egypt, and Iran [28,32,33,35]. The only study with the closest rates (EE 30%, DP 30%, and PA 25%) was that among oncology RDs and consultants in Egypt [26]. The higher rates in most studies could be because they involved only RDs (who are younger doctors and undergo more work stress), or were focused on one specialty with a greater tendency for burnout, such as emergency medicine.

For the role of socio-demographic factors, there is little discrepancy with other studies that employed multivariate analyses to ascertain independent predictors. This study's observation of significantly higher burnout amongst younger physicians mirrors the findings of others[26,27,35]. Unlike Fernando and Samaranayake [32], who observed higher risk among females, and Goyal and colleagues [49], who noted a stronger association with male sex, this study found no gender predisposition in keeping with the majority of other studies [15,18,25–28,32,33,35]. The unique cultural and systemic factors in Nigeria (and similar settings) could explain why gender does not emerge as a significant predictor of burnout. Generally, both male and female physicians face enormous societal expectations, although these might differ in form. For instance, while women may be expected to juggle professional and domestic responsibilities (given traditional gender roles around caregiving and home-making), men may experience the pressure of being the primary earners and decision-makers of their families. However, the distinction in these forms of stress may not be significant enough to differentiate the burnout experiences between the two genders, especially if both groups encounter similar systemic challenges such as high workloads and lack of resources in their professional life. Secondly, though the medical profession in Nigeria continues to be male-dominated[50], the expectations for physicians are the same, regardless of gender. Female physicians, particularly those entering the field in recent decades, are usually expected to adapt to their demanding roles just like their male counterparts.

Unlike in the study by Soltan et al [26], where there was no association between burnout and workload, this study showed manageability of workload as a negative correlate of EE, which is in agreement with most others where excessive perceived stress of work overload, higher occupational stress index, dealing with critical cases, more work hours, seeing more patients, and more frequent calls increased burnout[15,27,28,32,35,49]. Also, the observed negative linear relationship between burnout and reward at work supports other studies that found that burnout has an inverse relationship with

**Table 4. Association between Emotional Exhaustion (EE) and categorical independent variables.**

| Characteristic of Respondents | | Emotional Exhaustion (N=114) | | | | | | χ² | P-value |
|---|---|---|---|---|---|---|---|---|---|
| | | Low | | Moderate | | High | | | |
| | | n | % | n | % | n | % | | |
| Age (years) | ≤30 | 2 | 3.8 | 0 | 0.0 | 8 | 20.0 | 18.131 | *0.006 |
| | 31 – 40 | 23 | 43.4 | 10 | 47.6 | 22 | 55.0 | | |
| | 41 – 50 | 22 | 41.5 | 11 | 52.4 | 9 | 22.5 | | |
| | ≥ 51 | 6 | 11.3 | 0 | 0.0 | 1 | 2.5 | | |
| Sex | Male | 27 | 50.9 | 13 | 61.9 | 26 | 65.0 | 2.018 | 0.365 |
| | Female | 26 | 49.1 | 8 | 38.1 | 14 | 35.0 | | |
| Marital status | Single | 4 | 7.5 | 1 | 4.8 | 6 | 15.0 | 5.243 | 0.731 |
| | Married | 45 | 84.9 | 19 | 90.4 | 32 | 80.0 | | |
| | Separated | 2 | 3.8 | 0 | 0.0 | 1 | 2.5 | | |
| | Divorced | 1 | 1.9 | 1 | 4.8 | 0 | 0.0 | | |
| | Widow | 1 | 1.9 | 0 | 0.0 | 1 | 2.5 | | |
| Specialty category | Medical | 33 | 63.4 | 10 | 47.7 | 21 | 52.5 | 7.734 | 0.102 |
| | Surgical | 16 | 30.8 | 7 | 33.3 | 18 | 45.0 | | |
| | Diagnostic | 3 | 5.8 | 4 | 19.0 | 1 | 2.5 | | |
| Rank | Consultant | 14 | 26.4 | 3 | 14.3 | 5 | 12.5 | 8.415 | 0.078 |
| | Senior Registrar | 24 | 45.3 | 13 | 61.9 | 15 | 37.5 | | |
| | Registrar | 15 | 28.3 | 5 | 23.8 | 20 | 50.0 | | |
| Number of years post- graduation from medical school | ≤10 years | 16 | 30.2 | 4 | 19.0 | 20 | 50.0 | 15.602 | *0.016 |
| | 11 – 20 years | 24 | 45.3 | 16 | 76.2 | 18 | 45.0 | | |
| | 21 – 30 years | 12 | 22.6 | 1 | 4.8 | 2 | 5.0 | | |
| | ≥ 31 years | 1 | 1.9 | 0 | 0.0 | 0 | 0.0 | | |

χ² = Chi-squared test statistic |

*statistically significant

recognition from hospital managers, monthly salary [35], celebrating accomplishments, having enough money[18], and absence of economic problems and concerns about future career [28]. The absence of an association with control of work life and community is in line with a study that found that burnout is not affected by lack of control over office/training or poor cordiality with colleagues[18].

Regarding the COVID-19 context, though adequacy of resources and perception of risk showed a relationship with burnout on bivariate analysis, these turned out to be insignificant on logistic regression, like the study by Mahmood et al[18]. This is at variance with the finding elsewhere of an association between burnout and receipt of pandemic-related training, satisfaction with organizational infection control measures, having a colleague who is infected, and fear of the coronavirus infection [25]. The observed protective effect of good leadership is in keeping with other research that high-lighted the similar role of a supportive organizational and work environment [18,35].

In general, this study supports the general suggestion in the wider literature that, apart from age, rather than socio-demographic and personal professional variables (such as specialty and years of practice), the determinants of burnout are

**Table 5. Association between Depersonalization (DP) and categorical independent variables.**

| Characteristic of Respondents | | Depersonalization (N=114) | | | | | | χ² | P-value |
|---|---|---|---|---|---|---|---|---|---|
| | | Low | | Moderate | | High | | | |
| | | n | % | n | % | n | % | | |
| Age (years) | ≤30 | 2 | 2.6 | 3 | 13.0 | 5 | 33.3 | 16.482 | *0.011 |
| | 31 – 40 | 37 | 48.7 | 12 | 52.2 | 6 | 40.0 | | |
| | 41 – 50 | 32 | 42.1 | 7 | 30.4 | 3 | 20.0 | | |
| | ≥ 51 | 5 | 6.6 | 1 | 4.3 | 1 | 6.7 | | |
| | | | | | | | | | |
| Sex | Male | 44 | 57.9 | 14 | 60.9 | 8 | 53.3 | 0.212 | 0.900 |
| | Female | 32 | 42.1 | 9 | 39.1 | 7 | 46.7 | | |
| | | | | | | | | | |
| Marital status | Single | 5 | 6.6 | 3 | 20.0 | 3 | 20.0 | 10.190 | 0.252 |
| | Married | 67 | 88.2 | 19 | 66.7 | 10 | 66.7 | | |
| | Separated | 2 | 2.6 | 0 | 0.0 | 1 | 6.7 | | |
| | Divorced | 2 | 2.6 | 0 | 0.0 | 0 | 0.0 | | |
| | Widow | 0 | 0.0 | 1 | 4.3 | 1 | 6.7 | | |
| | | | | | | | | | |
| Specialty category | Medical | 45 | 60.0 | 12 | 52.2 | 7 | 46.7 | 1.277 | 0.865 |
| | Surgical | 25 | 33.3 | 9 | 39.1 | 7 | 46.7 | | |
| | Diagnostic | 5 | 6.7 | 2 | 8.7 | 1 | 6.6 | | |
| | | | | | | | | | |
| Rank | Consultant | 16 | 21.1 | 2 | 8.7 | 4 | 26.7 | 8.296 | 0.081 |
| | Senior Registrars | 39 | 51.3 | 10 | 43.5 | 3 | 20.0 | | |
| | Registrars | 21 | 27.6 | 11 | 47.8 | 8 | 53.3 | | |
| | | | | | | | | | |
| Number of years post- graduation from medical school | ≤10 years | 22 | 28.9 | 9 | 39.2 | 9 | 60.0 | 9.310 | 0.157 |
| | 11 – 20 years | 42 | 55.3 | 13 | 56.5 | 3 | 20.0 | | |
| | 21 – 30 years | 11 | 14.5 | 1 | 4.3 | 3 | 20.0 | | |
| | ≥ 31 years | 1 | 1.3 | 0 | 0.0 | 0 | 0.0 | | |

χ² = Chi-squared test statistic |
*statistically significant

essentially contextual factors related to organizational arrangements and management systems in the work setting[33]. The observed association between burnout and contextual factors (such as workload, reward, and leadership) contributes to a broader understanding of physician burnout in resource-constrained settings by highlighting how these elements interact to create an environment where burnout is more likely to occur and more difficult to mitigate. In Nigeria, for example, physicians continue to face high patient volumes, insufficient staffing, limited tools, and poor remuneration[51]. It is also acknowledged that healthcare professionals occupying leadership positions in both the public and private sectors often lack the leadership skills and competencies required for their roles[52]. Recognizing these patterns can lead to more effective strategies aimed at supporting physician well-being and improving patient care in such challenging settings.

## Strengths/limitations of the study and discussion of the research process

The strengths of this study include the use of renowned and previously validated instruments which assessed several potential predictors, and multivariate analyses to ascertain their independent association with the outcomes of interest. Only cross-sectional studies were included in the literature review for direct comparability.

**Table 6. Correlation between burnout scores and continuous independent variables.**

| Characteristic of Respondents | | Emotional Exhaustion | Depersonalization | Diminished Personal Accomplishment |
|---|---|---|---|---|
| Work hours per week | *r* | 0.231 | 0.186 | 0.057 |
| | *p*-value | *0.014 | *0.047 | 0.547 |
| Number of calls per month | *r* | 0.017 | 0.153 | -0.101 |
| | *p*-value | 0.856 | 0.105 | 0.286 |
| Manageability of workload# | *r* | -0.538 | -0.269 | -0.129 |
| | *p*-value | *<0.0001 | *0.004 | 0.173 |
| Control over work life# | *r* | -0.309 | -0.278 | -0.023 |
| | *p*-value | *0.001 | *0.003 | 0.808 |
| Reward for work# | *r* | -0.411 | -0.165 | -0.021 |
| | *p*-value | *<0.0001 | 0.080 | 0.822 |
| Community at workplace# | *r* | -0.280 | -0.180 | 0.013 |
| | *p*-value | *0.003 | 0.055 | 0.890 |
| Fairness at workplace# | *r* | -0.326 | -0.225 | 0.070 |
| | *p*-value | *<0.0001 | *0.016 | 0.459 |
| Workplace values# | *r* | -0.307 | -0.252 | 0.008 |
| | *p*-value | *0.001 | *0.007 | 0.936 |
| Impact of pandemic# | *r* | 0.107 | 0.011 | 0.037 |
| | *p*-value | 0.256 | 0.910 | 0.696 |
| Adequacy of resources# | *r* | -0.263 | -0.175 | 0.010 |
| | *p*-value | *0.005 | 0.063 | 0.912 |
| Perception of risk | *r* | 0.241 | 0.237 | -0.001 |
| | *p*-value | *0.010 | *0.011 | 0.989 |
| Leadership | *r* | -0.452 | -0.308 | 0.030 |
| | *p*-value | *<0.0001 | *0.001 | 0.749 |

#Perception and experience of Respondents | *r* = Pearson correlation coefficient |
*Statistically significant

None of the significant associations with DP observed on bivariate analysis was maintained after logistic regression. This may have been due to depreciation in degrees of freedom (and, hence, precision) because the variables included in the regression model were too many for the dataset. It has been suggested that the ratio of events to parameters in a regression model should be 10:1[53]. Therefore, with only 15 persons having high DP, having seven variables in the model was likely a pitfall. Another reason for the initially significant predictors becoming insignificant may be having two or more input variables that have a correlation, as is most likely the case for age and number of years post-graduation. The

**Table 7. Result of binary logistic regression model for factors associated with high Emotional Exhaustion.**

| Characteristic of respondents | | Adjusted odds ratio | 95% CI | | *P*-value |
|---|---|---|---|---|---|
| | | | Lower | Upper | |
| **Categorical variables** | **Age group** | | | | |
| | ≤30 years^ | | | | |
| | 31 − 40 years | 0.189 | 0.021 | 1.688 | 0.136 |
| | 41 − 50 years | 0.050 | 0.004 | 0.651 | *0.022 |
| | ≥ 51 years | 0.008 | 0.000 | 1.056 | 0.053 |
| | **Years post-graduation** | | | | |
| | ≤10 years^ | | | | |
| | 11 − 20 years | 0.810 | 0.189 | 3.469 | 0.776 |
| | 21 − 30 years | 3.249 | 0.146 | 72.486 | 0.457 |
| | ≥ 31 years | 0.000 | 0.000 | 0.000 | 1.000 |
| | | | | | |
| **Continuous variables** | Work hours per week | 1.001 | .986 | 1.017 | 0.862 |
| | Workload# | 0.094 | 0.027 | 0.328 | *0.000 |
| | Control# | 2.118 | 0.838 | 5.352 | 0.113 |
| | Reward# | 0.427 | 0.205 | 0.892 | *0.024 |
| | Community# | 0.799 | 0.343 | 1.861 | 0.602 |
| | Fairness# | 0.843 | 0.266 | 2.674 | 0.772 |
| | Workplace values# | 0.694 | 0.281 | 1.715 | 0.429 |
| | Adequacy of resources# | 1.018 | 0.841 | 1.231 | 0.856 |
| | Risk perception | 1.047 | 0.936 | 1.170 | 0.424 |
| | Leadership# | 0.525 | 0.113 | 0.929 | *0.043 |

#Perception and experience of respondents | ^Reference category | CI = Confidence Interval |

*statistically significant

consequence is that the effect of each input variable gets dispersed and attenuated [53]. It could have been better to use only one of the related parameters.

A serious limitation of this study is the low response rate with its implications for the external validity since the sample may be unrepresentative of the study population, as individuals who turn down research invitations tend to differ from those who participate [54]. The extent of this non-response bias could have been determined by comparing the characteristics of responders with the same characteristics of non-responders [55], but it was impracticable to obtain tangible information about non-respondents. The sample size was also below the calculated required minimum, which means that it had a reduced statistical power to detect statistically significant differences where such differences exist, and the survey estimates may lack precision[56]. Survey response rates have been on the decline in recent times due to progressively increasing time pressures on people, and medical doctors respond less than other health workers[56]. This may have affected this study, especially with the shortage of medical personnel due to the massive brain drain and the impact of the COVID-19 pandemic on health personnel and systems. Making repeated contact with selected study subjects is a strategy for improving recruitment [54], and this was employed in this study with three reminder WhatsApp messages (at roughly two-week intervals) after the initial contact. These were done courteously, always bearing in mind respect for autonomy [55].

One person replied that he started the survey but could not complete it because the questions triggered him. This suggests that the obtained prevalence of burnout is less than the true picture since non-response, even as low as 25%, distorts estimates of disease burden when the disorder itself is responsible for the non-response [54]. As a duty of care, the physician who experienced emotional distress was directed to organizations where he could get help. Many of the

doctors gave strong feedback that the questionnaire was too long. Indeed, the questionnaire had four sections with a total of 95 questions which, added to the detailed PIS and consent form, are burdensome, especially on a hand-held device. Lengthy questionnaires and online surveys are known to receive lower responses[56]. More practically, a shortened form of the MBI could have been used since abbreviated versions have been found to correlate strongly with their respective domain scores from the full MBI and give identical results in the estimation of associations [9].

### Recommendations

Addressing contextual factors related to organizational arrangements and management systems in the work setting, most importantly workload, leadership, and reward, is key to controlling burnout among RDs and consultants in Nigeria. In this regard, the following measures should be adopted by Nigerian policy makers and health system managers.

1. More emphasis should be placed on the mental health of graduate medical trainees and trainers. There should be an intervention to create more awareness of burnout among RDs and consultants through continuous training programs that incorporate coping strategies for overcoming work stress and preventing burnout. These should include time management, handling difficult clinical cases, satisfying home—work demands, mindfulness techniques, cognitive behavioral therapy, etc. Periodic screening as well as counselling systems should also be instituted[15,29,30,32].

2. Strategies for stress reduction and revision of workload distribution in tertiary hospitals are needed, e.g., a limit on the number of work hours per week, number of overnight calls per month, etc., and stipulating adequate breaks and vacations [15,49,57]. Health systems should be structured to make workload manageable by not placing excessively high demands on RDs, or stifling their autonomy at work [33,58].

3. There is a need for a paradigm shift from the traditional paternalistic form of medical leadership, which is based only on the merits of years of service, academic excellence, and professional competence with little or no emphasis on leadership capacity[59]. With the increasing complexity in medical practice and health systems, medical leadership in Nigeria should also evolve. Appointees to the positions of Chief Medical Director and Director of Clinical Services and Training (at the level of the hospital), Residency Program Coordinator (at the level of the department), and other management positions in healthcare organizations should be persons with vision and well-schooled in critical components of leadership including effective communication, engendering teamwork/supportive organizational culture, and giving adequate recognition to the accomplishments of subordinates [59,60]. "Professional staff not suited to leadership, either through training, selection or natural inclination, should not be entrusted with administrative and management burdens" if possible. [61]. The curricula for medical education, starting from the first professional degree in medicine, should be revised to promote training in medical leadership, prepare future managers, and enshrine the culture of leadership from the early stages of their careers [59]. This can be achieved by incorporating training modules in medical schools, residency training programs, and the methods of continuing medical education so that leadership is taught and reinforced across the continuum of medical training. The modules should cover leadership skills vital in clinical settings, such as communication (including listening), risk assessment, decision-making, empathy, provision of support, confidence, optimism, and transparency[62].

4. Mentorship programs should be incorporated into the residency training in all accredited health facilities, geared at giving younger doctors all the support they can get since they are more prone to burnout [63].

5. Adequate reward systems should be designed. Mechanisms should be instituted for the motivation of medical personnel to stimulate job enthusiasm so that they perform their duties with optimum interest and satisfaction in pursuit of organizational and personal goals[64]. Extrinsic motivation in the form of tangible and intangible incentives from managers of health systems and organizations, such as bonuses, recognitions, and salary increases, can be helpful in controlling burnout among RDs and consultants in Nigeria, especially against the backdrop of the current poor work conditions that encourage constant brain drain[21,65].

6. More research on the issue of burnout among RDs and consultants in Nigeria is recommended. Qualitative studies to give a better understanding of these physicians' experiences of burnout, and longitudinal studies to identify causal factors will support control efforts.

## Conclusion

This study examined the prevalence of and factors associated with burnout among RDs and consultants in the two tertiary hospitals in Delta State, Nigeria. It showed an overall high burnout of 40.4%, with significant predictors being age, workload, leadership, and reward. It suggests significant levels of burnout among a key component of the medical workforce in Nigeria whose mental health and productivity have grave implications for health care. It also reveals predictors that highlight critical areas for intervention in Nigerian healthcare organizations, especially those involved in residency training. Given the dire consequences of physician burnout on patient care, this study highlights the need for national healthcare policies and interventions that enhance the work life of physicians based on a framework of stress reduction, manageable workloads, effective leadership, and adequate rewards. This should be viewed as fundamental to making the best of health system performance, optimizing patient satisfaction, improving population health, and cutting healthcare costs [18,57,60].

Though this study offers valuable insights into the factors contributing to physician burnout in a Nigerian setting, the findings might also be adapted or applied to address burnout in similar resource-constrained locales of low- and middle-income countries by focusing on the specific contextual factors that could be exacerbating physician burnout in those environments and implementing tailored interventions.

## Supporting information

**S1 Data. Minimal data set.**
(SAV)

## Author contributions

**Conceptualization:** Nnamdi Stephen Moeteke, Ezinneamaka Erhirhie.

**Formal analysis:** Nnamdi Stephen Moeteke.

**Investigation:** Nnamdi Stephen Moeteke.

**Methodology:** Nnamdi Stephen Moeteke, Ezinneamaka Erhirhie.

**Project administration:** Nnamdi Stephen Moeteke, Ezinneamaka Erhirhie.

**Resources:** Nnamdi Stephen Moeteke, Ezinneamaka Erhirhie.

**Software:** Nnamdi Stephen Moeteke.

**Validation:** Ezinneamaka Erhirhie.

**Visualization:** Nnamdi Stephen Moeteke, Ezinneamaka Erhirhie.

**Writing – original draft:** Nnamdi Stephen Moeteke, Ezinneamaka Erhirhie.

**Writing – review & editing:** Nnamdi Stephen Moeteke, Ezinneamaka Erhirhie.

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
