## [Decision Letter · Decision Letter 0]

2 Jan 2025

PMEN-D-24-00416

Physician burnout in the context of the COVID-19 pandemic: prevalence and associated factors among resident doctors and consultants in Delta State, Nigeria

PLOS Mental Health

Dear Dr. Moeteke,

Thank you for submitting your manuscript to PLOS Mental Health. After careful consideration, we feel that it has merit but does not fully meet PLOS Mental Health’s publication criteria as it currently stands. Therefore, we invite you to submit a revised version of the manuscript that addresses the points raised during the review process. I apologise for the severe delay in this decision reaching you.

Please ensure that all reviewer points are fully addressed.

We look forward to receiving your revised manuscript.

Kind regards,

Karli Montague-Cardoso

Executive Editor

PLOS Mental Health

Journal Requirements:

1. When completing the data availability statement of the submission form, you indicated that you will make your data available on acceptance. We strongly recommend all authors decide on a data sharing plan before acceptance, as the process can be lengthy and hold up publication timelines. Please note that, though access restrictions are acceptable now, your entire data will need to be made freely accessible if your manuscript is accepted for publication. This policy applies to all data except where public deposition would breach compliance with the protocol approved by your research ethics board. If you are unable to adhere to our open data policy, please kindly revise your statement to explain your reasoning and we will seek the editor's input on an exemption. Please be assured that, once you have provided your new statement, the assessment of your exemption will not hold up the peer review process.

2. We do not publish any copyright or trademark symbols that usually accompany proprietary names, eg (R), (C), or TM  (e.g. next to drug or reagent names). Please remove all instances of trademark/copyright symbols throughout the text, including “Transform™ Survey Hosting”, “Maslach Burnout Toolkit™”, “Maslach Burnout Inventory™” on page 14.

Additional Editor Comments (if provided):

Reviewers' comments:

Reviewer's Responses to Questions

**Comments to the Author**

1. Does this manuscript meet PLOS Mental Health’s publication criteria ? Is the manuscript technically sound, and do the data support the conclusions? The manuscript must describe methodologically and ethically rigorous research with conclusions that are appropriately drawn based on the data presented.

Reviewer #1: Yes

Reviewer #2: Yes

2. Has the statistical analysis been performed appropriately and rigorously?

Reviewer #1: Yes

Reviewer #2: Yes

3. Have the authors made all data underlying the findings in their manuscript fully available (please refer to the Data Availability Statement at the start of the manuscript PDF file)?

Reviewer #1: Yes

Reviewer #2: No

4. Is the manuscript presented in an intelligible fashion and written in standard English?

Reviewer #1: Yes

Reviewer #2: Yes

5. Review Comments to the Author

Reviewer #1: Abstract

The abstract provides a well-structured overview of the study, including background, aims, methods, results, and conclusions. However, a few adjustments would enhance clarity and flow:

1. The first sentence could be clearer. Instead of "Residents Doctors (RDs) and consultants carry out the most specialised medical care," it may be more accurate to state: "Resident Doctors (RDs) and consultants provide specialized medical care." Also, consider introducing the term "burnout" earlier in the background to ensure it is immediately clear why the study is relevant.

2. The phrase "Previously validated instruments were used to collect data via an online survey" could benefit from mentioning the specific instruments used (e.g., Maslach Burnout Inventory). This would provide a more concrete understanding of the methodology.

3. The conclusion is strong, but it might be helpful to explicitly state the significance of the findings for future interventions in Nigerian healthcare systems.

Introduction

1. In the first paragraph, the definitions of "resident doctor" and "consultant" are useful but could be more concise. The current phrasing is somewhat repetitive with the following sentences. Try simplifying for brevity.

2. The burnout definition is clear, but it might be beneficial to emphasize its relevance to healthcare professionals earlier in the introduction. The mention of "incessant brain drain" and "unconducive conditions" is important, but the connection to burnout could be made more explicit here rather than later in the text.

3. Consider explaining briefly how the COVID-19 pandemic has exacerbated the challenges faced by physicians in the country. For example, the strain on healthcare systems due to COVID-19 could be mentioned in the context of these pre-existing issues.

4. While it is crucial to emphasize the role of workplace-related stressors, consider streamlining the points about "lack of essential tools/equipment, inefficient systems, poor remuneration, and absence of support" to a more concise list. Perhaps combining these under broader categories like "resource shortages" or "organizational challenges" could help improve flow.

5. The introduction does well to highlight the gap in research regarding physician burnout in low- and middle-income countries, especially during the COVID-19 pandemic. The comparison with other regions could be strengthened by mentioning some specific studies or data from similar contexts to establish a stronger foundation for the study's significance. For example https://www.frontiersin.org/journals/psychiatry/articles/10.3389/fpsyt.2024.1373743/full , https://journals.plos.org/plosone/article?id=10.1371/journal.pone.0309701 , https://bmjopen.bmj.com/content/11/9/e054284.abstract , https://journals.plos.org/plosone/article?id=10.1371/journal.pone.0305713

6. How do the research objectives build upon previous studies in Nigeria, and how does this study address gaps identified in the literature?

7. Could you elaborate on the rationale for focusing on tertiary hospitals in Delta State, Nigeria?

Methods

5. It would be helpful to briefly mention why a cross-sectional design was chosen, particularly in relation to the study objectives

6. Consider providing additional information on the characteristics of the facilities, such as the specific types of services they provide or their size. This could provide a better context for understanding the generalizability of the findings

7. It might be helpful to mention whether the study includes any specific characteristics or exclusions within the group of RDs and consultants (e.g., specialty, gender, etc.), if applicable.

8. In the multistage sampling technique, you might want to clarify how the participants were selected within the specialty and cadre groups (e.g., were participants randomly chosen, or was there another method used?).

9. Could you clarify why the specific instruments and scales used (e.g., MBI) were chosen over other potential tools?

10. Was there any pre-testing of the survey instrument to ensure its suitability for the study population?

11. Given the low response rate, did you perform any sensitivity analysis or adjustments to account for potential response bias?

12. It might be useful to briefly explain how the PEPS was adapted or validated for this particular study

13. The authors should include information on how the online format might have impacted participation or data quality (e.g., completion rates, dropout rates, etc.).

14. The authors should mention any steps taken to check for multicollinearity or other issues that could affect the validity of the regression model.

15. Consider briefly mentioning whether any steps were taken to protect participant confidentiality and anonymity beyond password protection (e.g., data storage procedures, who had access to the data, etc.).

16. Clarify if there were any incentives for participation or compensation for participants

Results

17. The multivariate analysis presents interesting findings, particularly with the likelihood of older age groups experiencing lower EE. However, the authors should clarify what other variables were controlled for in this analysis to strengthen the interpretation of the results. For instance, did the analysis control for years of experience or professional rank when assessing the effects of age on burnout?

Discussion

18. You mention that burnout levels in your study are lower than in certain other African countries. Could you clarify the key contextual differences that might explain this variance?

19. How do the observed associations between burnout and contextual factors such as workload and leadership contribute to the broader understanding of physician burnout in resource-constrained settings?

20. In the case of gender and burnout, you note no significant association. Could you provide more insight into potential cultural or systemic factors that might explain this finding in your context?

Limitations/Strengths

21. Regarding the sample size and low response rate, do you think the burnout levels reported might underestimate the true prevalence? How might this impact the generalizability of your findings?

22. Could you provide more detail on how you addressed emotional discomfort among participants, particularly in cases where distress might be linked to burnout itself?

Recommendations

23. The recommendations emphasize improving leadership capacity. Did your findings identify specific gaps in current leadership training or practices that need to be addressed?

24. You propose a revision of medical education curricula to include leadership training. Could you clarify how you envision this being implemented within the Nigerian medical education system?

Conclusion

25. Could you expand on how your findings might influence national healthcare policies, particularly regarding the brain drain issue and its connection to physician burnout?

26. How might the study's findings be adapted or applied to address burnout in other regions of Nigeria or similar low- and middle-income countries?

Reviewer #2: Data was collected between 13th January and 5th March 2022. Please add when the Pandemic began in Nigeria, when it peaked and when it waned. This is important as our experience in India was that duirng the first wave in 2020-21 in India there was widespread psychological distress but by end of 2021 and in 2022 the psychological distress had waned. In addition mention the average patient koad with COVID-19 in the tertiary care institute.

6. PLOS authors have the option to publish the peer review history of their article (what does this mean? ). If published, this will include your full peer review and any attached files.

**Do you want your identity to be public for this peer review?** For information about this choice, including consent withdrawal, please see our Privacy Policy .

Reviewer #1: **Yes: ** Amir Kabunga

Reviewer #2: **Yes: ** Dr. Suprakash Chaudhury

---

## [Editor Report · Decision Letter 1]

17 Mar 2025

Physician burnout in the context of the COVID-19 pandemic: prevalence and associated factors among resident doctors and consultants in Delta State, Nigeria

PMEN-D-24-00416R1

Dear Dr Moeteke,

We are pleased to inform you that your manuscript 'Physician burnout in the context of the COVID-19 pandemic: prevalence and associated factors among resident doctors and consultants in Delta State, Nigeria' has been provisionally accepted for publication in PLOS Mental Health.

Best regards,

Karli Montague-Cardoso

Executive Editor

PLOS Mental Health